# Deep Learning-Based Slice Thickness Reduction for Computer-Aided Detection of Lung Nodules in Thick-Slice CT

**DOI:** 10.3390/diagnostics14222558

**Published:** 2024-11-14

**Authors:** Jonghun Jeong, Doohyun Park, Jung-Hyun Kang, Myungsub Kim, Hwa-Young Kim, Woosuk Choi, Soo-Youn Ham

**Affiliations:** 1VUNO Inc., Seoul 06541, Republic of Korea; jonghun.jeong@vuno.co (J.J.); dhpark.ee@gmail.com (D.P.); junghyun.kang@vuno.co (J.-H.K.); 2Department of Applied Bioengineering, Graduate School of Convergence Science and Technology, Seoul National University, Seoul 08826, Republic of Korea; 3Department of Radiology, Kangbuk Samsung Hospital, Sungkyunkwan University School of Medicine, Seoul 03181, Republic of Korea; medpiano.kim@samsung.com (M.K.); choiws829@naver.com (W.C.); 4Department of Radiology, CHA Gangnam Medical Center, CHA University, Seoul 06125, Republic of Korea; hwayoungkim@chamc.co.kr

**Keywords:** deep learning, computer-aided detection, lung nodule, slice thickness reduction, computed tomography

## Abstract

Background/Objectives: Computer-aided detection (CAD) systems for lung nodule detection often face challenges with 5 mm computed tomography (CT) scans, leading to missed nodules. This study assessed the efficacy of a deep learning-based slice thickness reduction technique from 5 mm to 1 mm to enhance CAD performance. Methods: In this retrospective study, 687 chest CT scans were analyzed, including 355 with nodules and 332 without nodules. CAD performance was evaluated on nodules, to which all three radiologists agreed. Results: The slice thickness reduction technique significantly improved the area under the receiver operating characteristic curve (AUC) for scan-level analysis from 0.867 to 0.902, with a *p*-value < 0.001, and nodule-level sensitivity from 0.826 to 0.916 at two false positives per scan. Notably, the performance showed greater improvements on smaller nodules than larger nodules. Qualitative analysis confirmed that nodules mistaken for ground glass on 5 mm scans could be correctly identified as part-solid on the refined 1 mm CT, thereby improving the diagnostic capability. Conclusions: Applying a deep learning-based slice thickness reduction technique significantly enhances CAD performance in lung nodule detection on chest CT scans, supporting the clinical adoption of refined 1 mm CT scans for more accurate diagnoses.

## 1. Introduction

Lung cancer is one of the leading causes of cancer-related mortality globally, and early diagnosis is essential for better patient outcomes [1]. Computed tomography (CT) is widely used for screening and diagnosing lung diseases, with the detection of pulmonary nodules playing a pivotal role [2,3,4]. However, as the number of annual CT scans increases worldwide, manual interpretation by radiologists has become increasingly burdensome. It is also time-consuming and prone to inter-observer variability, particularly in detecting small or subtle nodules [5]. Intra-observer variability is also significant and can vary the interpretation of nodule types [6]. Recently, computer-aided detection (CAD) systems have been developed to assist radiologists in identifying lung nodules more efficiently and accurately [7].

Thin-slice CT (≤1.5 mm) is typically recommended for the accurate detection and measurement of pulmonary nodules because of its superior spatial resolution, which minimizes partial volume effects and enhances the visibility of small nodules [8]. High-resolution imaging is particularly important for detecting part-solid nodules, which have a higher probability of malignancy than other types of nodules [9]. Therefore, thick-slice CT (approximately 5.0 mm) is often used for screening to address these concerns. However, a reduced resolution can compromise the detection of small and subtle nodules [10]. This trade-off between image quality and radiation exposure poses a significant challenge for lung cancer screening and follow-up examinations [11]. However, in secondary hospitals or screening centers that primarily conduct lung cancer screenings, older CT scanners with thick-slice settings are often used due to the high costs associated with upgrading to thin-slice scanners. This means that not only is the trade-off between image resolution and radiation exposure significant, but it is also a practical problem in the clinical field.

To overcome these limitations, recent research has focused on applying deep learning-based slice thickness reduction (STR) techniques to enhance the resolution of thick-slice CT images [12,13]. These techniques can convert 3 or 5 mm CT scans into 1 mm scans, thereby improving the performance of CAD systems in the detection of lung nodules. The prior studies also demonstrate a decline in nodule detection performance with 5 mm slices. In Park et al.’s study [12], the sensitivity of the CAD system improved from 79.2% to 88.9% in 5 mm slice thickness—an increase of nearly 10%—after applying the STR technique. For reference, the sensitivity of 1 mm slice thickness CT scans was 92.0%, demonstrating already high performance. Building on these findings, our study confirms that STR enhances CAD performance using clinical data collected from screening centers in tertiary hospitals in Korea that perform routine lung cancer screenings. Therefore, as explained in Section 2, we used nearly twice the amount of data compared to previous studies for validation. Moreover, to align with practical clinical constraints, only 5 mm slice thickness CT scans were collected and used for validation. However, an inherent limitation is that the ground-truth (GT) nodules were also annotated solely from 5 mm slice thickness CT images. Additionally, applying an STR technique to thick-slice CT images significantly increases the sensitivity of CAD systems in detecting subsolid nodules, particularly non-solid nodules, which are more challenging to identify owing to their low contrast [14]. These enhanced images allow for better visualization of small and low-contrast nodules that are often missed in thicker slices owing to partial volume effects [15].

The purpose of this study was to assess the efficacy of an STR technique and a commercial CAD system that reduces the slice thickness from 5 mm to 1 mm and automatically detects lung nodules.

## 2. Materials and Methods

### 2.1. Study Design

#### 2.1.1. Patient Population

This study was conducted with the approval of the Institutional Review Board of Kangbuk Samsung Medical Center (KBSMC) (IRB File No. KBSMC 2021-01-065-007). As shown in Figure 1, we retrospectively collected 970 CT scans obtained at the KBSMC between January 2017 and December 2019. All patients were aged ≥19 years and underwent low-dose axial chest CT scans with 5 mm slice thickness, each covering at least 250 mm in the axial direction.

According to the Picture Archiving and Communication System (PACS, Seoul, Republic of Korea), there were 638 scans with nodules and 332 normal scans without nodules. We randomly selected 500 scans with nodules as dictated by the research contract. The nodules were annotated by three radiologists (Soo-Youn Ham, Hwa-Young Kim, and Myungsub Kim, with 32, 17, and 8 years of experience, respectively). In this study, we analyzed the performance of the two nodule sets. Nodule set A consists of 1078 nodules identified in 463 abnormal scans out of the 500 selected scans, where at least two out of three radiologists agreed on the identification of each nodule. Nodule set B included 535 nodules found in 355 abnormal scans out of the 500 selected scans, and all three radiologists agreed on the identification of each Berlin, Germanynodule. The remaining 37 scans in set A and 145 scans in set B were excluded from each analysis. The 332 normal scans were collected based on reports from the PACS, and Prof. Soo-Youn Ham, one of the lead radiologists involved in nodule annotation, verified each case to be non-nodular.

#### 2.1.2. CT Protocol

Low-dose chest CT scans were obtained with the following acquisition parameters: 110–140 kVp, 60–165 mA/s, and an interval of 5 mm with 5 mm slice thickness, and pixel spacing ranging from 0.5 mm to 0.9 mm.

The study population for the 832 scans analyzed by radiologists (500 scans with nodules and 332 scans without nodules) was distributed as 556 scans from males and 140 from females; unfortunately, gender information was unavailable for 136 scans (these scans are labeled as ‘others’ in Table 1). Regarding the scanner manufacturers, 680 scans were acquired using GE MEDICAL SYSTEMS (Milwaukee, WI, USA), and 152 scans were acquired using SIEMENS (Berlin, Germany) scanners, meaning all 832 scans were obtained using devices from these two manufacturers. Table 1 below provides a more detailed breakdown of these characteristics, separated by scans by gender and presence of nodules.

#### 2.1.3. Slice Thickness Reduction (STR)

The STR technique used to convert 5 mm slice thickness CT scans into 1 mm scans was implemented using a commercial CAD system (VUNO Med LungCT-AI; VUNO Inc., Seoul, Republic of Korea, https://www.vuno.co/ [accessed on 26 August 2024]), which utilized a convolutional neural network-based deep learning model. For a more comprehensive explanation of the methodology and validation results, please refer to a previous study [12,13]. In this study, we defined the 5 mm slice thickness CT scans as original 5 mm CT and 1 mm slice thickness CT scans generated using the STR technique as refined 1 mm CT.

The deep learning model applied in the STR technique was designed with a CNN-based super-resolution (SR) algorithm that employed residual learning to achieve slice thickness reduction. The STR model was trained with 88 CT datasets (divided based on factors such as radiation dose and use of contrast enhancement) and validated with 12 separate datasets, following the pipeline established in previous studies [12]. The training pipeline involved image pairs of thick-slice (3 mm or 5 mm) and thin-slice (1 mm) CT scans, where the algorithm learned to map low-resolution input images to high-resolution outputs.

In prior studies [12,13], STR models were developed and validated separately for 3 mm and 5 mm slice thicknesses. However, in our study, only 5 mm slice thickness CT images were collected, so we used only the 5 mm STR model. Data augmentation techniques, such as rotation and flipping, were applied to increase robustness, and the model was optimized using mean squared error (MSE) as the loss function, the adaptive moment estimation (ADAM) optimizer, and a learning rate of 0.0001.

#### 2.1.4. Computer-Aided Detection

Lung nodules were detected using a commercial CAD system (VUNO Med LungCT-AI). The CAD system was applied to both the original 5 mm CT and the refined 1 mm CT scans. VUNO Med LungCT-AI provides a segmentation mask for each detected nodule, along with a confidence score indicating the likelihood that the object within the region of interest (ROI) is a nodule. It also automatically measures the major and minor axes of the nodule on the plane (axial, coronal, or sagittal) with the largest segmentation mask, using these as representative axes and considering their intersection as the center point of the detected nodule.

#### 2.1.5. Matching Criteria

In the detection task, accurately identifying an object requires consideration of both the localization of the ROI and the corresponding confidence score [16]. For instance, even if the ROI perfectly overlaps the bounding box of the ground-truth object, it cannot be accurately detected if the confidence score is below the decision threshold. Similarly, a high confidence score is insufficient if it corresponds to an area far from the ground-truth object, resulting in a false positive. Therefore, it is necessary to establish matching criteria between the results obtained from the CAD system and ground-truth nodules. For example, in the Lung Nodule Analysis 2016 (LUNA16) challenge, a CAD mark is considered a true positive if its center point lies within a distance *r* from the center of any nodule in the ground-truth dataset, where *r* is the radius of the ground-truth nodule [7].

In this study, radiologists annotated each nodule by drawing the major and minor axes on the axial slice with the largest nodule size, as shown in Figure 2. The diameter of the nodule was defined as the average length of the major and minor axes, and the center point of the nodule was defined as the intersection of these axes. Annotations from different radiologists were considered to refer to the same nodule if the distance between the two centers was less than 1.5 times the average of the radii of both nodules. The final ground-truth center point and diameter were determined by averaging the center points and diameters of all radiologists. We considered a nodule detected by the CAD system to be correctly localized (though not yet a true positive, as the decision depends on the confidence score threshold) if its center point was located within 1.5 times the radius of the ground-truth nodule, calculated using the x, y, and z coordinates. All matched items from the CAD system had continuous confidence scores ranging from 0 to 1, which were used for further statistical analysis.

#### 2.1.6. Statistical Analysis

To evaluate the performance of the CAD system for lung nodule detection in CT scans, the following metrics were used: For per-scan analysis, we employed receiver operating characteristic (ROC) curves and area under the ROC curves (AUCs). In cases without nodules, the highest confidence score for any non-lesion localization was defined as the scan-level probability. Similarly, in cases with nodules, the scan-level probability was determined by the highest confidence score from either the lesion or non-lesion localization. For per-nodule analysis, we employed free-response ROC (FROC) curves, which plot the number of false positives per scan on the horizontal axis and nodule-level sensitivity on the vertical axis, comprehensively evaluating the detection performance by considering all false positives and all ground-truth nodules. We conducted subgroup analyses based on nodule size for both per-scan and per-nodule evaluations. According to the Lung CT Screening Reporting and Data System (Lung-RADS) guidelines, many categories are differentiated based on a nodule size of 6 mm and nodule type (solid, part-solid, non-solid, etc.) [17]. The malignancy risk increases progressively from Category 1 to Category 4X, and as the risk level rises, the recommended management approach becomes more urgent and varied. When categorizing nodules, the risk level also differs based on the nodule type—whether it is solid, part-solid, or non-solid. Notably, part-solid nodules are often classified into higher-risk categories than solid nodules, even when they are smaller in size. Additionally, the 6 mm size threshold is frequently used as a critical criterion in distinguishing categories by nodule type.

In the quantitative analysis, we will conduct a subgroup analysis by dividing nodules based on this 6 mm size threshold, enabling us to compare the effectiveness of our STR technique across larger and smaller nodule groups. The *p*-values for comparing AUCs were calculated using DeLong’s method [18]. Statistical significance was set as *p* < 0.05. All statistical analyses were performed using R version 4.3.2 (R Core Team (2020), R: A language and environment for statistical computing, R Foundation for Statistical Computing, https://www.R-project.org/ [accessed on 26 August 2024]).

For nodule-type assessment, we plan to perform a qualitative analysis to compare morphological features directly. In thick-slice CT images, part-solid nodules often appear blurred due to the thick axial projection, causing the solid component to blend and sometimes resemble a ground-glass nodule (GGN). This phenomenon, also described in previous studies [13], is what we previously referred to as the partial volume effect. Accordingly, in this study, we aim to assess, through qualitative analysis, whether applying our STR technique restores these blurred solid parts, potentially reclassifying GGNs as part-solid nodules.

## 3. Results

### 3.1. Quantitative Analysis

#### 3.1.1. Per-Scan Analysis

For nodule set A, the AUC of the CAD system on the refined 1 mm CT was significantly higher than that on the original 5 mm CT, with a value of 0.879 (95% confidence interval [CI], 0.856–0.902) compared to 0.850 (95% CI, 0.824–0.875), with a *p*-value of 0.002. For nodule set B, the AUC of the CAD system on the refined 1 mm CT was also significantly higher than that on the original 5 mm CT, with a value of 0.902 (95% CI, 0.881–0.922) compared to 0.867 (95% CI, 0.843–0.892), with a *p*-value of <0.001. The ROC curves and more details are shown in Figure 3a,c and Table 2 and Table 3.

According to the subgroup analysis based on nodule size, the STR technique was more effective for smaller nodules. For nodule set A, in cases where the nodules had a ground-truth diameter of less than 6 mm (947 out of 1078 nodules, and 427 out of 463 abnormal scans), the AUC of the CAD system on refined 1 mm CT was significantly higher than that on the original 5 mm CT, with a value of 0.860 (95% CI, 0.835–0.885) compared to 0.823 (95% CI, 0.794–0.851), with a *p*-value of <0.001. For cases with nodules 6 mm or larger in diameter (131 out of 1078 nodules, and 109 out of 463 abnormal scans), the AUC of the CAD system on refined 1 mm CT was also higher than that on the original 5 mm CT, with a value of 0.921 (95% CI, 0.888–0.954) compared to 0.891 (95% CI, 0.851–0.932), but this difference was not statistically significant, with a *p*-value of 0.126. For nodule set B, in cases where the nodules had a ground-truth diameter of less than 6 mm (447 out of 535 nodules, and 301 out of 355 abnormal scans), the AUC of the CAD system on refined 1 mm CT was significantly higher than that on the original 5 mm CT, with a value of 0.880 (95% CI, 0.857–0.904) compared to 0.844 (95% CI, 0.816–0.872), with a *p*-value of < 0.001. For cases with nodules 6 mm or larger in diameter (88 out of 535 nodules, and 82 out of 355 abnormal scans), the AUC of the CAD system on refined 1 mm CT was also significantly higher than that of the original 5 mm CT, with a value of 0.972 (95% CI, 0.959–0.985) compared to 0.946 (95% CI, 0.914–0.978), with a *p*-value of 0.049. The ROC curves for each subgroup are shown in Figure 3b,d.

#### 3.1.2. Per-Nodule Analysis

For nodule set A, 463 cases with 1078 nodules were analyzed. The sensitivities of the CAD system at 0.125, 0.25, 0.5, 1, 2, and 4 false positives per scan were 0.368, 0.487, 0.625, 0.720, 0.784, and 0.854 for refined 1 mm CT, and 0.328, 0.414, 0.520, 0.615, 0.675, and 0.704 for the original 5 mm CT, respectively. The competition performance metric (CPM), reflecting the average sensitivity, was 0.640 for the refined 1 mm CT and 0.543 for the original 5 mm CT. For nodule set B, 355 cases with 535 nodules were analyzed. The sensitivities of the CAD system at 0.125, 0.25, 0.5, 1, 2, and 4 false positives per scan were 0.555, 0.677, 0.789, 0.854, 0.916, and 0.953 for refined 1 mm CT, and 0.503, 0.581, 0.690, 0.779, 0.826, and 0.854 on the original 5 mm CT, respectively. The CPM was 0.791 for the refined 1 mm CT and 0.706 for the original 5 mm CT. The FROC curves and results are shown in Figure 4a,c and Table 4 and Table 5.

Similarly to the per-scan analysis, the nodule size-based subgroup analysis suggested that the STR technique was more effective for smaller nodules. For nodule set A, in cases with nodules with a ground-truth diameter of less than 6 mm (947 out of 1078 nodules), the sensitivities of the CAD system at 0.125, 0.25, 0.5, 1, 2, and 4 false positives per scan were 0.320, 0.439, 0.591, 0.697, 0.767, and 0.845 on refined 1 mm CT and 0.283, 0.366, 0.483, 0.589, 0.653, and 0.682 on the original 5 mm CT, respectively. The CPM was 0.610 for the refined 1 mm CT and 0.509 for the original 5 mm CT in this subgroup. For cases with nodules with a ground-truth diameter of 6 mm or larger (131 out of 1078 nodules), the sensitivities of the CAD system at 0.125, 0.25, 0.5, 1, 2, and 4 false positives per scan were 0.702, 0.802, 0.847, 0.893, 0.908, and 0.924 for refined 1 mm CT and 0.649, 0.756, 0.802, 0.817, 0.847, and 0.863 on the original 5 mm CT, respectively. The CPM in this subgroup was 0.846 for the refined 1 mm CT and 0.789 for the original 5 mm CT. For nodule set B, in cases with nodules with a ground-truth diameter of less than 6 mm (447 out of 535 nodules), the sensitivities of the CAD system at 0.125, 0.25, 0.5, 1, 2, and 4 false positives per scan were 0.492, 0.629, 0.754, 0.828, 0.899, and 0.946 for refined 1 mm CT and 0.436, 0.519, 0.647, 0.747, 0.803, and 0.839 on the original 5 mm CT, respectively. The CPM was 0.758 for the refined 1 mm CT and 0.665 for the original 5 mm CT in this subgroup. In contrast, or cases with nodules with a ground-truth diameter of 6 mm or larger (88 out of 535 nodules), the sensitivities of the CAD system at 0.125, 0.25, 0.5, 1, 2, and 4 false positives per scan were 0.886, 0.955, 0.977, 0.989, 0.989, and 0.989 for refined 1 mm CT and 0.830, 0.898, 0.920, 0.932, 0.932, and 0.932 on the original 5 mm CT, respectively. The CPM in this subgroup was 0.964 for the refined 1 mm CT and 0.907 for the original 5 mm CT. The FROC curves are shown in Figure 4b,d, respectively.

### 3.2. Qualitative Analysis

Figure 5a illustrates a nodule that was faintly visible on the original 5 mm CT scan but became much more clearly detectable on the refined 1 mm CT scan. The improvement in visibility is not simply due to the detection of a small nodule, but also because the nodule’s Hounsfield unit (HU) values increased during the STR, resulting in enhanced contrast. Figure 5b shows a ground-glass nodule that was detected on the original 5 mm CT scan but not on the refined 1 mm CT scan. During the STR process, the nodule became adjacent to the vessel, which may have caused confusion in its detection. Figure 5c presents nodules that appear as ground-glass nodules on the original 5 mm CT but are classified as part-solid nodules on the refined 1 mm CT. Part-solid nodules have a higher malignancy rate than ground-glass nodules, which makes them clinically more important. The STR technique can accurately categorize nodule types that may be misclassified in thick-slice CT scans.

## 4. Discussion

This study demonstrates that utilizing a deep learning-based STR technique to convert original 5 mm CT scans into refined 1 mm CT scans significantly enhances the performance of CAD systems in detecting lung nodules, particularly smaller nodules. Additionally, this technique accurately categorizes nodules that could be mistaken for ground-glass nodules as more clinically significant part-solid nodules. These findings are crucial for improving the early detection of lung cancer and patient outcomes, especially in clinical settings where thick-slice CT scans are commonly used.

Thin-slice CT scans (≤1.5 mm) are utilized in various clinical applications, particularly in the detection and evaluation of pulmonary nodules and lung cancer [8]. Studies have shown that thin-slice CT performs better in detecting nodules, which are more challenging to assess with thicker-slice CT, mainly because of partial volume effects [19,20]. Despite the advantages of thin-slice CT, there are clinical needs and benefits of thick-slice CT [10]. For instance, in routine follow-up examinations or screening programs, thicker slices help reduce the radiation dose and minimize the cumulative radiation exposure over time. This approach is safer for patients requiring regular monitoring. In pediatric or vulnerable populations, reducing radiation exposure is a priority, and thicker slices can offer acceptable image quality while adhering to the principle of As Low As Reasonably Achievable (ALARA) to minimize the radiation dose [21]. Additionally, in many lung cancer screening centers, practical constraints often lead to using older CT scanners that can only perform thick-slice (≥5 mm) imaging due to cost-related limitations. In some cases, even when thin-slice CT capabilities are available, a shortage of radiologists may necessitate thick-slice CT imaging for efficiency, as thin-slice CT generates a larger number of slices. Our STR technique could address these practical challenges by enhancing diagnostic accuracy and efficiency in lung cancer screening initiatives across many countries.

However, many studies have reported decreased nodule detection performance when thick-slice CT is used [22,23,24]. To overcome this problem, we utilized the STR technique to convert the original 5 mm CT images into refined 1 mm CT images. This approach significantly improved detection performance, demonstrating the enhanced ability of the CAD system to differentiate between cases with and without nodules. With the same specificity of nearly 0.80, the accuracy and sensitivity also showed marked improvements with the STR technique. Our analysis revealed that STR was effective for nodules both smaller and larger than 6 mm in diameter, with the benefits being particularly pronounced for nodules less than 6 mm. This indicates that these smaller and more challenging nodules became more discernible with enhanced resolution in the axial direction. With nodule set A, the performance for nodules larger than 6 mm improved, but the difference between the original 5 mm and refined 1 mm CT scans was not statistically significant. In contrast, in nodule set B, which included more confidently identified nodules, CAD performance showed a statistically significant improvement with the STR technique for larger nodules. In particular, with 5 mm CT images, vessels may appear interrupted or fragmented due to the thicker slice thickness, potentially leading to misinterpretation as nodules smaller than 6 mm (resulting in false positives). We believe that our STR technique could help improve this issue by enhancing the reconstructed continuity of vascular structures in the images.

Another important discovery is that nodules appearing as ground glass on the original 5 mm CT scans can be identified as part-solid on refined 1 mm CT scans, a result consistent with findings from previous studies [13]. This differentiation is critical, as part-solid nodules have a higher likelihood of malignancy and require prompt treatment [17]. However, one issue is that we do not have paired data obtained with 5 mm and 1 mm slice thicknesses, that is, ground-truth data from 1 mm CT, making it uncertain whether the nodule is actually part-solid. Nonetheless, such changes in nodule characteristics highlight the importance of thin-slice CT and STR technology as complementary approaches. Additionally, we observed that the STR technique effectively reduced the partial volume effect commonly observed in thick-slice CT. It is likely that some low-attenuation and smaller nodules were not visible or easily characterized owing to this effect, which underscores the importance of reviewing CAD marks on thin slices [10]. Therefore, minimizing the partial volume effect is another advantage of the STR technique. These results indicate the benefits of STR techniques in clinical decision-making.

Nonetheless, this study had several limitations. First, this was a single-center study, which may limit the generalizability of the findings to other settings or populations. Second, because no ground-truth 1 mm CT scan was available, the performance of the STR technique could not be independently validated, potentially introducing experimental bias. As a result, qualitative analysis had to be performed based on the refined 1 mm CT scans, which might not fully represent the actual characteristics of the nodules. Future studies should aim to confirm these results in more diverse cohorts and thoroughly analyze the performance of STR techniques.

## 5. Conclusions

This study highlights the potential benefits of using STR techniques to improve the CAD system performance in detecting lung nodules. By enhancing the image resolution in the axial direction, these techniques can help to reduce missed diagnoses, particularly for small nodules. This resulted in a significant improvement in the detection performance, with the AUC increasing from 0.867 to 0.902. Additionally, the STR technique provides a more accurate characterization of nodule types and improves the overall image quality. Nevertheless, this study requires further validation across multiple centers, and the reliability of the images generated by the STR technique should be verified before attributing diagnostic performance improvements to the method. In summary, this study validated STR techniques that may contribute to better early screening for lung cancer and improve patient care.

## Figures and Tables

**Figure 1 diagnostics-14-02558-f001:**
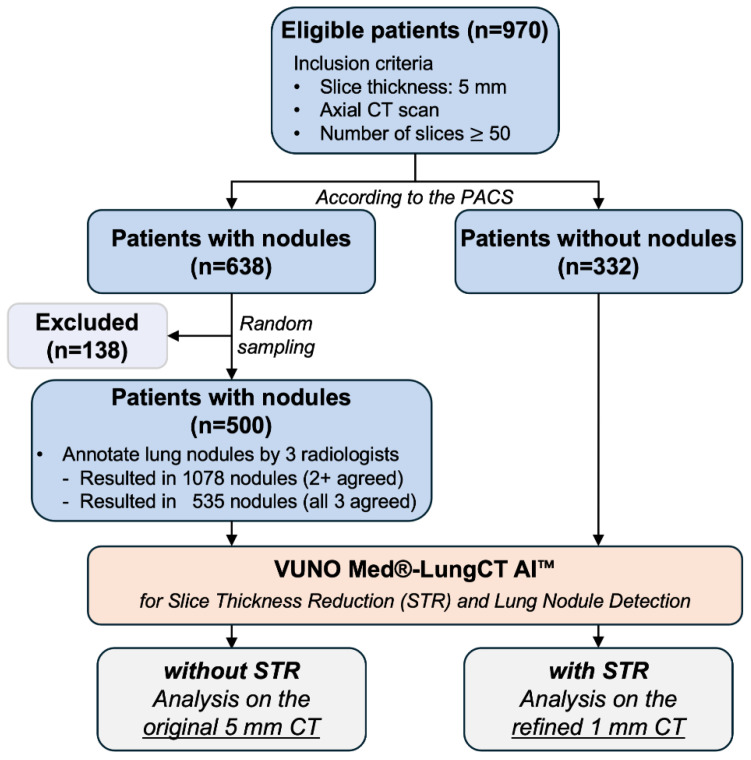
Data flow diagram. This study compares and analyzes the performance of the computer-aided detection system on original 5 mm CT scans and refined 1 mm CT scans obtained through a slice thickness reduction technique.

**Figure 2 diagnostics-14-02558-f002:**
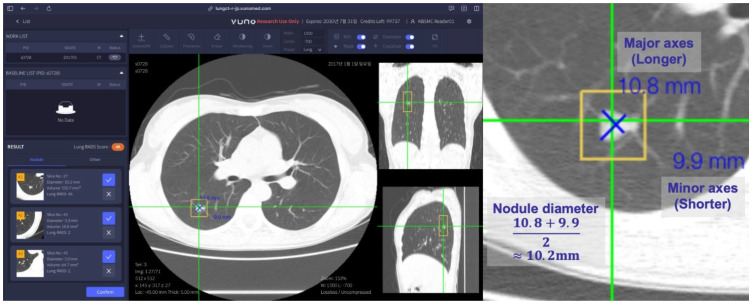
Example images of a lung nodule and its annotations. The left image was captured from commercially available software (VUNO-Med LungCT-AI; ver. VN-M-04), and the right image is a magnified example.

**Figure 3 diagnostics-14-02558-f003:**
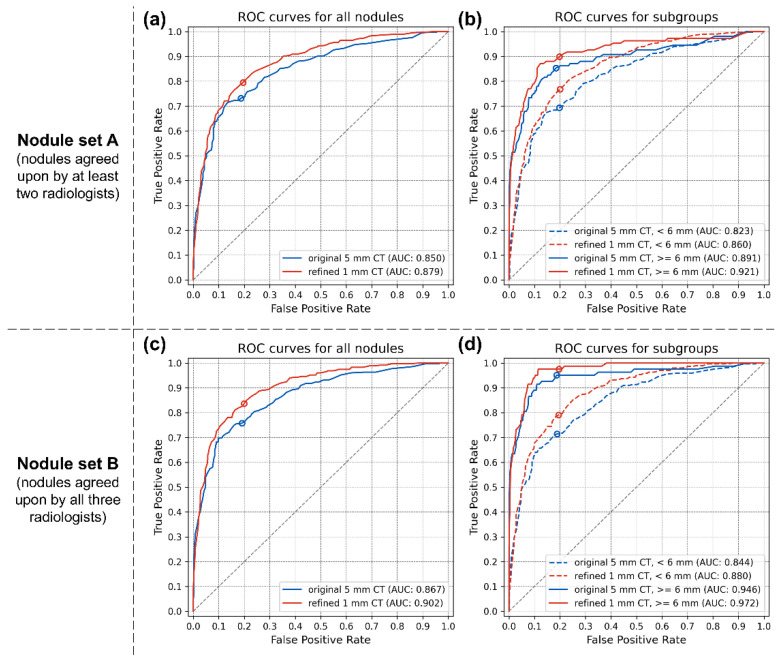
ROC curves for per-scan analysis. (**a**) Nodule set A without subgroup analysis. (**b**) Nodule set A with subgroup analysis based on nodule size. (**c**) Nodule set B without subgroup analysis. (**d**) Nodule set B with subgroup analysis based on nodule size. The blue and red lines represent CAD results from the original 5 mm and refined 1 mm CT scans, respectively. In the subgroup analyses, dashed and solid lines represent CAD results for nodules smaller than 6 mm and those 6 mm or larger, respectively. ROC, receiver operating characteristic; AUC, area under the ROC curve.

**Figure 4 diagnostics-14-02558-f004:**
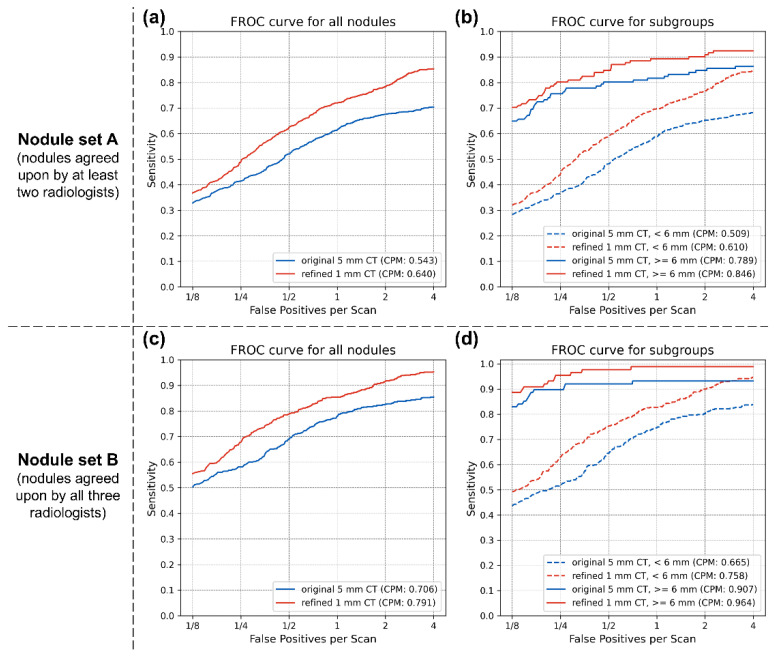
FROC curves for per-nodule analysis. (**a**) Nodule set A without subgroup analysis. (**b**) Nodule set A with subgroup analysis based on nodule size. (**c**) Nodule set B without subgroup analysis. (**d**) Nodule set B with subgroup analysis based on nodule size. The red and blue lines represent CAD results from the original 5 mm and refined 1 mm CT scans, respectively. In the subgroup analyses, solid and dashed lines represent CAD results for nodules 6 mm or larger and those smaller than 6 mm, respectively. FROC, free-response receiver operating characteristic; CAD, computer-aided detection.

**Figure 5 diagnostics-14-02558-f005:**
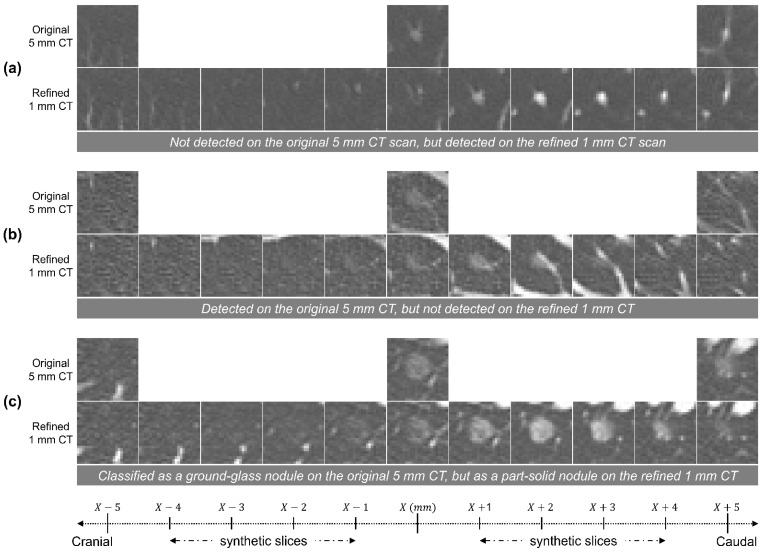
Representative images for the quantitative analysis between the original 5 mm CT scan and the refined 1 mm CT scan. (**a**) Not detected on the original 5 mm CT scan but detected on the refined 1 mm CT scan. (**b**) Detected on the original 5 mm CT but not detected on the refined 1 mm CT. (**c**) Classified as a ground-glass nodule on the original 5 mm CT but as a part-solid nodule on the refined 1 mm CT. The cranial direction is toward the left, while the caudal direction is toward the right. The original 5 mm CT images were obtained at 5 mm intervals, while the refined 1 mm CT images were obtained at 1 mm intervals.

**Table 1 diagnostics-14-02558-t001:** Dataset demographics and study population.

Characteristic	Patients with Nodules	Patients Without Nodules
Analyzed Scans *	500	332
Sex		
Male	273	283
Female	91	49
Others *	136	
Manufacturer		
GE MEDICAL SYSTEMS	387	293
SIMENSE	113	39
Annotated Nodule Scans		
2+ radiologists agreed	463	-
All 3 radiologists agreed	355	-

“Analyzed Scans *” refers to the scans that radiologists reviewed for nodule annotation. The “Others *” category in the gender row does not indicate individuals of a different gender; rather, it represents scans for which gender information was missing.

**Table 2 diagnostics-14-02558-t002:** Computer-aided detection (CAD) result for per-scan analysis with nodule set A, which are nodules agreed upon by at least two radiologists.

Image	AUC (95% CI)	*p*-Value	Accuracy	Sensitivity	Specificity
All nodules					
original 5 mm CT scan	0.850 (0.824–0.875)	-	0.768	0.732	0.813
refined 1 mm CT scan	0.879 (0.856–0.902)	0.002	0.799	0.795	0.805
Nodules with <6 mm					
original 5 mm CT scan	0.823 (0.794–0.851)	-	0.746	0.693	0.802
refined 1 mm CT scan	0.860 (0.835–0.885)	<0.001	0.784	0.768	0.800
Nodules with ≥6 mm					
original 5 mm CT scan	0.891 (0.851–0.932)	-	0.821	0.853	0.816
refined 1 mm CT scan	0.921 (0.888–0.954)	0.126	0.815	0.899	0.802

Confidence intervals (CIs) were calculated using DeLong’s method. AUC, area under the receiver operating characteristic curve.

**Table 3 diagnostics-14-02558-t003:** Computer-aided detection (CAD) result for per-scan analysis with nodule set B, which are nodules agreed upon by all three radiologists.

Image	AUC (95% CI)	*p*-Value	Accuracy	Sensitivity	Specificity
All nodules					
original 5 mm CT scan	0.867 (0.843–0.892)	-	0.787	0.758	0.809
refined 1 mm CT scan	0.902 (0.881–0.922)	<0.001	0.816	0.837	0.801
Nodules with <6 mm					
original 5 mm CT scan	0.844 (0.816–0.872)	-	0.776	0.714	0.812
refined 1 mm CT scan	0.880 (0.857–0.904)	<0.001	0.800	0.791	0.806
Nodules with ≥6 mm					
original 5 mm CT scan	0.946 (0.914–0.978)	-	0.827	0.951	0.813
refined 1 mm CT scan	0.972 (0.959–0.985)	0.049	0.821	0.976	0.804

Confidence intervals (CIs) were calculated using DeLong’s method. AUC, area under the receiver operating characteristics curve.

**Table 4 diagnostics-14-02558-t004:** Computer-aided detection (CAD) result for per-nodule analysis with nodule set A, which are nodules agreed upon by at least two radiologists.

Image	Sensitivity at Specific FPs per Scan						
	1/8	1/4	1/2	1	2	4	CPM (Average)
All nodules (*n* = 1078)							
original 5 mm CT scan	0.328	0.414	0.520	0.615	0.675	0.704	0.543
refined 1 mm CT scan	0.368	0.487	0.625	0.720	0.784	0.854	0.640
	(+0.040)	(+0.073)	(+0.105)	(+0.105)	(+0.109)	(+0.150)	(+0.097)
Nodules with <6 mm (*n* = 947)							
original 5 mm CT scan	0.283	0.366	0.483	0.589	0.653	0.682	0.509
refined 1 mm CT scan	0.320	0.439	0.591	0.697	0.767	0.845	0.610
	(+0.037)	(+0.073)	(+0.108)	(+0.108)	(+0.114)	(+0.163)	(+0.101)
Nodules with ≥6 mm (*n* = 131)							
original 5 mm CT scan	0.649	0.756	0.802	0.817	0.847	0.863	0.789
refined 1 mm CT scan	0.702	0.802	0.847	0.893	0.908	0.924	0.846
	(+0.053)	(+0.046)	(+0.045)	(+0.076)	(+0.061)	(+0.061)	(+0.057)

*n* represents the number of nodules. FP, false positive; CPM, competition performance metric.

**Table 5 diagnostics-14-02558-t005:** Computer-aided detection (CAD) result for per-nodule analysis with nodule set B, which are nodules agreed upon by all three radiologists.

Image	Sensitivity at Specific FPs per Scan						
	1/8	1/4	1/2	1	2	4	CPM (Average)
All nodules (*n* = 535)							
original 5 mm CT scan	0.503	0.581	0.690	0.779	0.826	0.854	0.706
refined 1 mm CT scan	0.555	0.677	0.789	0.854	0.916	0.953	0.791
	(+0.052)	(+0.096)	(+0.099)	(+0.075)	(+0.090)	(+0.099)	(+0.085)
Nodules with <6 mm (*n* = 447)							
original 5 mm CT scan	0.436	0.519	0.647	0.747	0.803	0.839	0.665
refined 1 mm CT scan	0.492	0.629	0.754	0.828	0.899	0.946	0.758
	(+0.056)	(+0.110)	(+0.107)	(+0.081)	(+0.096)	(+0.107)	(+0.093)
Nodules with ≥6 mm (*n* = 85)							
original 5 mm CT scan	0.830	0.898	0.920	0.932	0.932	0.932	0.907
refined 1 mm CT scan	0.886	0.955	0.977	0.989	0.989	0.989	0.964
	(+0.056)	(+0.057)	(+0.057)	(+0.057)	(+0.057)	(+0.057)	(+0.057)

*n* represents the number of nodules. FP, false positive; CPM, competition performance metric.

## Data Availability

Data are contained within the article.

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
