# Peer review of "Deep Learning-Based Slice Thickness Reduction for Computer-Aided Detection of Lung Nodules in Thick-Slice CT"

_diagnostics, 2024, doi:10.3390/diagnostics14222558_

Round 1

Reviewer 1 Report

Comments and Suggestions for Authors

I examined your work titled "Deep Learning-based Slice Thickness Reduction for Computer-Aided Detection of Lung Nodules in Thick Slice CT" in detail. The paper presents a slice thickness reduction technique on CT scans to improve the performance of CAD systems. I want to point out that the study is written well in general terms. I listed a few minor points I found missing in the study in bullet points:

1- The introduction of the article is not well organized. The contribution of the paper to the literature should be discussed in detail and a few items should be added at the end of the section. The innovative aspects of the proposed approach should be emphasized. The shortcomings of existing studies in the field of improving CAD performance as well as the highlights of this study should be highlighted.

2-There is no related studies section in the article. The deficiencies of these studies should be emphasized by including the existing published studies in the literature and a comprehensive comparison should be made with the proposed method by adding the results obtained from these studies to the experimental results section. The difference of this study should be emphasized by making comparisons with other modern methods used in terms of CAD performance.

3- The 687 CT scans used in the study were evaluated retrospectively, but the process of creating the dataset and how the scans were selected are not detailed enough. It would be appropriate to add a table detailing the dataset in the patient population section. If the dataset is publicly available, a link or citation regarding access should be considered. The selection criteria for non-nodule scans are not clear.

4- The deep learning-based slice thickness reduction technique used in the study should be further detailed. Details such as the CT protocol, the type of model used in STR technique approaches, its training, and how it was optimized should be clearly stated. The technical details of the applied method and how it was applied should be included.

5- Results are presented with statistical metrics such as AUC, accuracy, and sensitivity, but the methodology of the analyzed statistics and the significance of the measurable results are not sufficiently addressed. A more descriptive assessment can be made in the Qualitative Analysis section.

6- It should be addressed how the study results will contribute to clinical practice. In this regard, concrete suggestions should be made on how the proposed Slice thickness reduction technique can be used in practice.

7- The article emphasizes that the method provides greater improvement in small nodules, and the leading reasons for this performance increase should be explained in more detail.

8- The Conclusions section is insufficient. This section should include more detailed evaluation and highlight the effectiveness of the method. According to the 2 limitations given above, it has been stated that the generalizability of the method to other settings or populations may be limited and since no real 1-mm CT scan is available, the performance of the STR technique could not be independently validated. It seems that these limitations will compromise the validity of the proposed study. It would be useful to include more detailed explanations of the limitations.

Author Response

Comment 1: The introduction of the article is not well organized. The contribution of the paper to the literature should be discussed in detail and a few items should be added at the end of the section. The innovative aspects of the proposed approach should be emphasized. The shortcomings of existing studies in the field of improving CAD performance as well as the highlights of this study should be highlighted.

Response 1: Thank you for your invaluable feedback. In response to your suggestions, we have revised the Introduction to better emphasize the contributions and innovation of our study. Firstly, we added citations to highlight the importance of AI-based CAD systems in lung cancer screening and additional references on how slice thickness variations impact radiomics features and diagnostic performance.

We also clarified that prior studies reported limited CAD performance with 5-mm slice thickness CT. Building upon these findings, we collected routine lung cancer screening CT data in a Korean tertiary hospital and demonstrated that CAD performance improves after applying STR, specifically showing that STR can significantly aid in detecting subsolid nodules that are challenging to identify in thick-slice CT images. This addition enhances the novelty and relevance of our study.

Thank you once again for your valuable feedback, which has dramatically helped improve the clarity and impact of our manuscript.

Comment 2: There is no related studies section in the article. The deficiencies of these studies should be emphasized by including the existing published studies in the literature and a comprehensive comparison should be made with the proposed method by adding the results obtained from these studies to the experimental results section. The difference of this study should be emphasized by making comparisons with other modern methods used in terms of CAD performance.

Response 2: We fully agree with your suggestion to include a 'related studies' section to compare prior research with ours and highlight our study's innovation. However, as some of this content is already discussed in the Introduction, we have chosen to expand the Introduction rather than add a separate related studies section.

In response, we have added a more detailed summary of the primary prior study, Park et al. [12], and clarified that our study differentiates itself by using clinical data that aligns more closely with real-world clinical settings (* lung cancer screening), with a larger number of cases collected for validation. We also disclosed the limitations of our study, specifically that we collected only 5 mm slice thickness CT data and annotated ground truth (GT) nodules from these thick-slice CT images, too.

Thank you again for your feedback, which helped us refine our study's unique contributions and limitations and enhance the clarity of our manuscript.

Comment 3: The 687 CT scans used in the study were evaluated retrospectively, but the process of creating the dataset and how the scans were selected are not detailed enough. It would be appropriate to add a table detailing the dataset in the patient population section. If the dataset is publicly available, a link or citation regarding access should be considered. The selection criteria for non-nodule scans are not clear.

Response 3: We agree with your suggestion to provide more detailed information about the non-nodule scans and the study population. In response, we have made the following revisions:

  1. In Section 2.1.1 Patient Population, we explained the selection process for the non-nodule scans. Specifically, the 332 non-nodule scans were collected based on PACS reports, and their non-nodular status was verified by one of the lead radiologists (Prof. Soo-Youn Ham), who was involved in the nodule annotation process. We believe this verification by an experienced radiologist enhances the reliability.
  2. In Section 2.1.2 CT Protocol, we incorporated additional details about the study population, such as gender distribution and the manufacturers of the CT scanners. We also included Table 1 to summarize these characteristics, breaking down the information by scans with and without nodules.

Unfortunately, due to data privacy regulations and the fact that Kangbuk Samsung Hospital owns the dataset, it is not publicly accessible. However, per your recommendations, we have provided as much detailed information as possible to support reproducibility and transparency. Thank you again for your valuable feedback, which has helped us enhance the clarity of our dataset description.

Comment 4: The deep learning-based slice thickness reduction technique used in the study should be further detailed. Details such as the CT protocol, the type of model used in STR technique approaches, its training, and how it was optimized should be clearly stated. The technical details of the applied method and how it was applied should be included.

Response 4: Thank you for your insightful question. We would like to clarify that Park et al., 2019 thoroughly described the STR model details. Since this study used a previously validated model from prior studies [11,12] without additional training, we initially provided only a brief description in the manuscript. However, in line with your suggestion, we agree that a concise yet comprehensive explanation of the main text can further enhance readers' understanding.

Therefore, as requested, we have added more details regarding the STR model architecture, training pipeline, and optimized parameters in Section 2.1.3, Slice Thickness Reduction (STR). We believe this addition will improve clarity for readers. Thank you again for your valuable feedback.

Comment 5: Results are presented with statistical metrics such as AUC, accuracy, and sensitivity, but the methodology of the analyzed statistics and the significance of the measurable results are not sufficiently addressed. A more descriptive assessment can be made in the Qualitative Analysis section.

Response 5: In response to your suggestion, we have clarified the clinical significance of Lung-RADS in lung cancer screening, as well as the quantitative and qualitative evaluation methods, in Section 2.1.5 (Statistical Analysis). This section now provides a more detailed explanation of our approach to both quantitative and qualitative assessments, with specific attention to Lung-RADS guideline. Furthermore, in the Results section, we have thoroughly presented both quantitative (nodule size subgroup analysis) and qualitative findings (nodule type change) to demonstrate the improvements achieved through our STR technique. We have also highlighted the clinical implications of these findings in the Discussion section, providing additional context on the value of our approach in clinical settings.

Comment 6: It should be addressed how the study results will contribute to clinical practice. In this regard, concrete suggestions should be made on how the proposed Slice thickness reduction technique can be used in practice.

Response 6: In response, we have expanded the Discussion section to clarify the potential contributions of our STR technique to clinical practice. We first addressed the benefits of thick-slice CT in reducing radiation dose, especially for patients requiring regular monitoring and in pediatric and vulnerable populations. Following this, we discussed how our STR technique can enhance clinical practice by improving diagnostic accuracy and efficiency in settings where only thick-slice CT scanners are available or where thin-slice imaging may be impractical due to resource limitations. Regarding the practical challenges mentioned, we could not locate studies that provide national or international statistics on the prevalence of thick-slice CT scanners specifically used in lung cancer screening centers. Therefore, this explanation is based on feedback from clinical sites currently using our products.

Comment 7: The article emphasizes that the method provides greater improvement in small nodules, and the leading reasons for this performance increase should be explained in more detail.
Response 7: Thank you for your feedback. In response, we have clarified the reasons for the improved performance of our STR technique, particularly for smaller nodules, in the Discussion section. We attribute this to three main factors:

  1. The STR technique mitigates partial volume effects seen in thick-slice CT.
  2. The enhanced axial resolution allows smaller nodules to be more discernible.
  3. STR reduces false positives by reconstructing continuity in vascular structures, which may otherwise appear fragmented and be misinterpreted as nodules on 5-mm CT images.

Comment 8: The Conclusions section is insufficient. This section should include more detailed evaluation and highlight the effectiveness of the method. According to the 2 limitations given above, it has been stated that the generalizability of the method to other settings or populations may be limited and since no real 1-mm CT scan is available, the performance of the STR technique could not be independently validated. It seems that these limitations will compromise the validity of the proposed study. It would be useful to include more detailed explanations of the limitations.

Response 8: Thank you for your feedback. We have addressed your suggestions in the Conclusion section by emphasizing the effectiveness of our STR technique and clarifying the study's limitations. Please review these updates.

Reviewer 2 Report

Comments and Suggestions for Authors

- The manuscript describes a pipeline for CT slice thickness reduction in the context  of lung nodule detection; the overall results are encouraging, but the recommendation is to carefully address several aspects on the manuscript

- The first suggestion is to include an Abstract with the structure usually adopted in teh literature;

- Section 1 is found to be very limited in what concerns framing the research problem addressed, especially when there is not Background / Related Work section; a suggestion is to herein  include the arguments and references to previous works that are later presented in section 4;

- Inter-observer variability in medical image analysis is indeed  a relevant aspect (line 42), but that also goes for intra-observer fluctuations as it has been recently pointed out (eg:  doi: 10.1007/s10278-023-00800-5);

- In section 2, it should be clarified the number of images used (not only the number of nodules);

- Is it relevant to present a section (2.1.2) with only one sentence ? 

- An effort should be made to present in a more clear way sections Sections 2.1.4  section 2.1.5;

- The paper would benefit with a benchmark of the results against other methods.

Author Response

Comment 1: The first suggestion is to include an Abstract with the structure usually adopted in teh literature;

Response 1: The editor requested that we format the abstract using the structured style commonly employed in medical research papers, and we have revised it accordingly.

Comment 2: Section 1 is found to be very limited in what concerns framing the research problem addressed, especially when there is not Background / Related Work section; a suggestion is to herein include the arguments and references to previous works that are later presented in section 4;

Response 2: Thank you for the suggestion. While we did not add a separate Related Work section, we have expanded the Introduction to focus on relevant prior studies. Please review these updates

Comment 3: Inter-observer variability in medical image analysis is indeed a relevant aspect (line 42), but that also goes for intra-observer fluctuations as it has been recently pointed out (eg: doi: 10.1007/s10278-023-00800-5);

Response 3: Thank you for the suggestion. We have added a reference to emphasize the importance of intra-observer variability in diagnosing lung nodules and included a brief mention in the Introduction session to highlight its relevance.

Comment 4: In section 2, it should be clarified the number of images used (not only the number of nodules);

Response 4: We have reported both the number of scans (n=XXX) and the number of nodules.

Comment 5: Is it relevant to present a section (2.1.2) with only one sentence ?

Response 5: We added additional study population information and summarized it in Table 1.

Comment 6: An effort should be made to present in a more clear way sections Sections 2.1.4 section 2.1.5;

Response 6: We have revised and clarified both Sections 2.1.4 and 2.1.5 for improved clarity.

Comment 7: The paper would benefit with a benchmark of the results against other methods.

Response 7: In the introduction, we discuss the results of previous studies while highlighting the improvements demonstrated in our study. However, to the best of our knowledge, no other studies have developed alternative algorithms to directly benchmark the performance of STR in enhancing lung nodule CAD systems.

Reviewer 3 Report

Comments and Suggestions for Authors

Comments on the Quality of English Language

The text needs minor revisions in the English language.

Author Response

Comment 1. The introduction is very general. An analysis of specific methods used in the indicated area is lacking.

Response 1: Thank you for your feedback! We have expanded the Introduction to include a result summary of previous studies and have also mentioned the limitations of our research. Please review these updates.

Comment 2. What dictated the need for this research

Response 2: In the Introduction, we explain that the need for this research is driven by the continued, frequent use of thick-slice CT scans (5 mm slice thickness) in clinical practice. Such scans are prone to reduced lung nodule detection performance due to issues like the partial volume effect, which obscures smaller nodules or nodule types. Our study addresses this gap by proposing an STR technique to enhance the diagnostic accuracy of lung nodule detection in thick-slice CT images.

Comment 3. Is the database used available online? Can the authors provide a link to the database?

Response 3: Unfortunately, the CT data used in this study are not publicly available.

Comment 4: Database used is from 5 years ago. Does this in any way affect the research done by the authors? What is the quality of the data, today the images (tomographic scans) would have to be made with a higher quality and this would affect the research methods and, accordingly, the results obtained. According to the reviewer, the images in Figure 2 and Figure 5 are not clear enough.

Response 4: The images shown in Figures 2 and 5 were uploaded at 300 dpi, so the quality itself is clear. However, the 5-mm slice thickness inherently results in a slightly blurred appearance. The data used in this study were indeed collected 5 years ago, which contributed to a longer analysis time. These data were gathered in tertiary hospitals, and while screening protocols using 5-mm slice thickness are becoming less common, thick-slice scanners are still in use in clinical settings where lung cancer screenings are routinely conducted, as updated in the Introduction and Discussion sections.

Comment 5. The reviewer believes that the methods used in the research should be described in detail.

Response 5: Thank you for the suggestion. We have added details on the collection of non-nodule scans in Section 2.1.1 (Patient Population) and provided additional information in Section 2.1.2 (CT Protocol), along with Table 1. We also expanded Section 2.1.3 with a more detailed explanation of the STR algorithm. Additionally, in Section 2.1.6 (Statistical Analysis), we explained how quantitative and qualitative assessments were conducted from a clinical perspective. These updates have helped to make our manuscript more precise and rigorous.

Comment 6. It is good for the authors to specify and describe their contributions in this study.

Response 6: Thank you for the suggestion. This study's main contribution is validating the STR technique and analyzing its impact on CAD performance. If you are referring to individual author contributions, we have outlined each author’s specific contributions to the research and manuscript in the "Author Contributions" section at the end of the paper. Please let us know if further clarification is needed.

Comment 7. The conclusions should be supported by the results obtained from the performed analyses.

Response 7: Thank you. We have revised the Conclusion to include a summary of our performance presented in the Results section.

Comment 8. Are there more modern versions of the analysis software used? Or other software from recent years. Accuracy of results is very important in studies like this.

Response 8: In this study, we used the latest version of VUNO Med-LungCT available at the time of analysis (v1.0.0.18). A new updated version has been released as of the current revision.

Round 2

Reviewer 2 Report

Comments and Suggestions for Authors

The authors fairly addressed my previous comments and suggestions